# Evaluation of the relationship between consumption of carbonated soft drinks/fast food and anxiety-related sleep disturbance in school adolescents in Bangladesh

**Raufun Hasan Arnob, Shamima Akter, Md. Mosfequr Rahman** ⓘ *

Department of Population Science and Human Resource Development, University of Rajshahi, Rajshahi, Bangladesh

* mosfeque@ru.ac.bd

## Abstract

While studies from high-income countries have shown an association between adolescents' poor dietary habits and a lack of quality sleep, there is a dearth of similar data from developing nations. This study intends to investigate the relationship between the consumption of carbonated soft drinks and fast food and sleep disturbances linked to anxiety in school-going adolescents in Bangladesh. The data used for this study came from the 2014 Bangladesh Global School-based Health Survey. Information of 1746 adolescents was utilized in this current analysis. Multivariable logistic regression analyses were used to identify the associations of interest. In this sample, sleep disturbance associated with anxiety was prevalent at 3.5%. Approximately half of the adolescents (44.4%) consumed soft drinks for one or more occurrences per day during the past 30 days, and 51.2% consumed fast food on one or more days during the past 7 days. Results show that the odds of sleep disturbance associated with anxiety were higher among adolescents who consumed soft drinks (odds ratio [OR] = 2.43; 95% confidence interval [CI] = 1.15–5.15) and fast food (OR = 2.34; 95% CI = 1.01–5.43) than their respective counterparts after controlling for other covariates, such as age, gender, grade, feeling hungry, engagement in physical violence, physical activity, being bullied, having close friends, peer support, and parental attachment. Sleep disturbance due to anxiety is more common among Bangladeshi school-aged adolescents who consume carbonated beverages or fast food. Further longitudinal studies are necessary to validate or refute our findings and investigate relevant explanations.

## Introduction

High-quality sleep is crucial for the health of adolescents and has a significant role in their mental, physical, and emotional development [1]. Nevertheless, prior studies suggest that a significant number of adolescents have inadequate and low-quality sleep [2], which has been linked to a range of emotional and behavioral issues [3,4] and may also negatively impact their physical development. Globally, 36% of adolescents have been shown to encounter sleep issues

**Data availability statement:** The data for the current study are publicly available at the World Health Organization NCD Microdata Repository (https://extranet.who.int/ncdsmicrodata/index.php/catalog/485)

**Funding:** The authors have declared that no competing interests exist.

**Competing interests:** The authors have declared that no competing interests exist.

[5]. A recent systematic study and meta-analysis indicated that the total pooled prevalence of sleep problems among Chinese adolescents is 26% [6]. Another cohort study in Germany involving adolescents aged 10–17 years revealed that over 20% of individuals experienced sleep-related issues [7]. However, a multi-country study utilizing the most recent data from the Global School-Based Health Survey indicated that the pooled weighted prevalence of anxiety-induced sleep disturbance was 10.7% [8].

Inadequate sleep and sleep deprivation can adversely affect the motor skills, immunological function, attention, and academic performance of adolescents. Additionally, it may also raise the likelihood of experiencing suicidal thoughts and engaging in substance misuse [9,10]. The prevalence of weight gain and obesity in adolescents [11], which is a well-known risk factor for cardio-metabolic illnesses such as diabetes and cardiovascular disease in adulthood [12], may be attributed to the metabolic effects of inadequate sleep. Therefore, gathering evidence on the factors that can contribute to and are linked to sleep deprivation in adolescence can help promote mental health and physical well-being.

Recent research indicates that unhealthy eating habits, such as consuming high-calorie foods that are rich in carbohydrates and drinking carbonated soft drinks, are linked to sleep deprivation, sleep problems, and low sleep quality in children and adolescents [13–15]. A study conducted by Khan et al. [16] analyzed data from the Global School-based Student Health Survey and found that among adolescents, consuming carbonated soft drinks three or more times per day and fast foods four or more days per week was strongly linked to sleep disturbance. This association was observed in all countries except for low-income countries and was significant for both males and females across different regions defined by the World Health Organization. A study conducted in the United States among adolescents found that inadequate sleep (less than 10 hours per day) was linked to a higher frequency of consuming carbonated soft drinks, such as soda. However, no association was found between insufficient sleep and the consumption of juices or other sugar-sweetened beverages, such as sports drinks or fruit-flavored drinks [14]. A further study conducted with South Korean adolescents found that the perception of sleep satisfaction was strongly linked to the use of fast foods and carbonated soft beverages, resulting in a reduced likelihood [17].

The majority of these studies, however, are restricted to high-income countries, and there is a scarcity of literature on these findings in low-income nations. Bangladesh is currently experiencing a rapid expansion of its economy, which has the potential to bring about significant transformations in the lifestyles of adolescents. Their food habits have undergone a significant change. Easy accessibility and affordability of fast foods and sweetened beverages have led to increased consumption among Bangladeshi adolescents [18,19]. Moreover, historically it is evident that industrialization in high-income countries resulted in rural-to-urban migration and a shift in nutrition from predominantly plant-based foods—such as vegetables, fruits, whole grains, and legumes—to hyper-caloric diets characterized by high levels of total and saturated fats, cholesterol, animal protein, added salt, and sugar, while being low in fiber [20,21]. Currently, various regions globally, including Bangladesh, are undergoing rapid urbanization, which may lead to socio-economic changes as well as shifts in lifestyle and dietary patterns from traditional diets to Western-like diets [22]. This transition is accompanied by increased access to a wider variety of foods, including high-calorie and processed options [23]. The impact of lifestyle and dietary changes resulting from rapid urbanization in low- and middle-income countries, including Bangladesh, has accelerated the rising burden of non-communicable diseases.

In Bangladesh, approximately 20% of the population consists of adolescents aged 10–19 [24]. A study found that 11.6% of males and 4.9% of females reported consuming crispy or fried snacks (SCFS) at least 7 times per week. Additionally, 28.9% of males and 24.8% of

females reported consuming sugary snacks (SS) at least 7 times per week, while 25.6% of males and 20.7% of females reported consuming sugar-sweetened beverages (SSB) at least 7 times per week [25]. According to another survey, 13.4% of people consumed soft drinks three or more times per day, while 29.6% consumed fast food three or more days per week [26]. Adolescents who attend school are more frequently consuming fast foods and carbonated soft drinks, which can lead to an elevated risk of sleep disturbances and associated health problems. This, in turn, leads to higher healthcare expenses. There is a lack of research on the consumption of fast food and carbonated soft drinks, as well as the potential danger of sleep disturbance associated with anxiety among adolescents in Bangladesh. It is crucial to address this area of research in Bangladesh, where resources to meet adolescents' mental health needs are scarce, and there is evidence that the consumption of fast food and soft drinks is rising among young generations [27], in order to develop effective intervention strategies. Hence, the aim of this study was to evaluate the correlation between the consumption of carbonated soft drinks and fast food and the occurrence of sleep deprivation related to anxiety in adolescents attending school in Bangladesh.

## Methods

This study analyzed publicly available secondary data from the Global School-based Student Health Survey (GSHS), which was designed to collect data from school-aged adolescents (typically aged 11–17 years) in developing countries, including Bangladesh, and was administered by the World Health Organization (WHO) in collaboration with the Centers for Disease Control and Prevention [28]. The GSHS collects data on several aspects of adolescent health, such as eating habits (e.g., consumption of fast food and carbonated soft drinks) and well-being (e.g., anxiety-related sleep disturbance). Data were collected in GSHS using a standardized scientific sample selection approach, a traditional school-based methodology, a combination of core questionnaire modules, and expanded and country-specific questionnaires. The 2014 Bangladesh GSHS used a cross-sectional, two-stage cluster sampling design to select a representative sample from all students enrolled in grades 7, 8, 9, and 10. Initially, schools were selected using probability proportional to size sampling. Classes were selected at random from these schools, and all students in the chosen classes were included in the sampling frame and eligible to participate. Participants filled out a self-administered questionnaire and wrote their responses on a computer-scannable answer sheet. The school response rate in the 2014 Bangladesh GSHS was 97%, with 94% for students, for an overall response rate of 91%. The Ministry of Health and Family Welfare in Dhaka, Bangladesh, approved this study. Parents and guardians of schoolchildren were informed about the survey's purpose and content, their children's privacy, and the voluntary nature of participation. Only adolescents and their parents or guardians who provided written or verbal consent participated in the GSHS. Because the current study used retrospective, de-identified, publicly available data, no ethics approval was necessary for this secondary analysis. Information on various sociodemographic, behavioral, and health-related factors was collected from 2989 students. Deleting the case-wise missing values, this current analysis included a sample of 1746 adolescents. Full survey explanations are accessible at: https://extranet.who.int/ncdsmicrodata/index.php/catalog/485.

## Outcome

This study focuses on anxiety-induced sleep disturbance. This was assessed in the GSHS using the following question: "During the past 12 months, how often have you been so worried about something that you could not sleep at night?" The survey included response choices such as 'never,' 'rarely,' 'sometimes,' 'most of the time,' and 'always.' According to previous

research [16,29], individuals who answered 'most of the time' or 'always' were classified as experiencing sleep disturbance due to anxiety.

## Exposures

The frequency of carbonated soft drink consumption was evaluated using the following question: "During the past 30 days, how many times per day did you usually drink carbonated soft drinks, such as Coke, Fanta, Orange, or 7-UP?" The variable was classified as either 0 (no occurrence or fewer than once per day) or 1 (one or more occurrences per day). The frequency of fast food consumption was evaluated using the following question: "During the past 7 days, on how many days did you eat food from a fast food restaurant, such as KFC, BFC, or Pizza Hut?" For modeling purposes, this variable was classified as follows: 0 (indicating no consumption on any days) and 1 (indicating consumption on one or more days).

## Covariates

Based on the existing literature [30–36], the current analysis incorporates a set of covariates associated with sleep problems among adolescents, including age, gender, grade, food insecurity, experiences of bullying, presence of close friends, lack of peer support, parental attachment, engagement in physical activities, and exposure to physical violence. According to prior literature [37,38], the quantification of physical violence and physical activity involved aggregating the responses from two questions for each of them, whereas parental attachment was assessed by summing the responses from three items. Upon summing all elements within each variable, a result of 0 signifies "no," while a sum over 0 denotes "yes." A complete list of covariates used in this current analysis, with their categories and coding, is presented in Table 1.

## Statistical analysis

The χ2 test was used to identify differences in anxiety-related sleep disturbance based on the use of carbonated soft drinks and fast food, as well as other variables. The findings display the proportion of students who responded to each question, while any missing data were excluded from the study. Bivariate logistic regression analyses were utilized to assess the relationship between anxiety-induced sleep deprivation and the consumption of fast food, carbonated soft drinks, neither, either, or both. Multivariate logistic regression analyses assessed the relationship between anxiety-related sleep disturbances and the three primary exposure variables, controlling for all other covariates. The strength of the association was assessed using odds ratios, which were accompanied by 95% confidence intervals (CIs) for significance testing. Sample weights were employed to ensure that the GSHS data accurately reflected the population of Bangladesh. All the analyses were done with the svy estimation commands in STATA 14 MP (Stata Corp., College Station, TX). This was done to get accurate variance estimates and take into account the complicated sample design of the survey data. The statistical inferences were made at a significance threshold of 0.05.

## Results

Table 2 presents the demographic characteristics of the participants alongside the frequency of sleep disturbance attributed to anxiety. The majority of respondents were male, comprising 64.1%, while 10.1% of the students reported feelings of hunger. Approximately 45.8% of students reported involvement in physical violence, 88.5% participated in physical activities, and 24.5% indicated experiences of being bullied. Approximately 18.6% of students reported receiving support from peers, while 78.4% indicated strong relationships with their parents.

**Table 1.  The complete list of covariates included in this study.**

| Variables | Survey question | Coding |
|---|---|---|
| Age | How old are you? | 1=11–16 years<br>2= 16–18 years |
| Gender | What is your sex? | 1 = Male<br>2 = Female |
| Grade | In what class are you? | 1 = Class VII<br>2 = Class VIII<br>3 = Class IX<br>4 = Class X |
| Food insecurity | How often did you go hungry because there was not enough food in your home? | 0, no = Never/Rarely/Sometimes<br>1, yes = Most of the time/Always |
| Bullied | How many days you were bullied? | 0, no = Never<br>1, yes = One or more days |
| Physical Violence | During the past 12 months, how many times were you in a physical fight? | 0 = No<br>1 = Yes |
|  | During the past 12 months, how many times were you seriously injured? |  |
| Close friend | How many close friends do you have? | 0, no= No close friend<br>1, yes= Have at least one close friend |
| Peer support | During the past 30 days, how often were most of the students in your school kind and helpful? | 0 = No<br>1 = Yes |
| Parental attachment | How often did your parents check to see if your home-work was done? | 0, no= Never/rarely/sometimes<br>1, yes= Most of the time/always |
|  | How often did your parents or guardians understand your problems and worries? |  |
|  | How often your parents or guardians really know about what you were doing with your free time? |  |
| Physical activity | During the past 7 days, on how many days were you physically active for a total of at least 60 minutes per day? | 0, No = Did not physical activity<br>1, Yes = One or more times per week |
|  | During the past 7 days, on how many days did you walk or ride a bicycle to or from school? |  |

The occurrence of anxiety-related sleep disturbance within the student sample was 3.5% (Table 2).

The prevalence rate of soft drink consumption was 44.4%, whereas fast food consumption exhibited a prevalence rate of 51.2% (Fig 1). Fig 2 illustrates the prevalence of anxiety-related sleep disturbances in relation to the consumption of soft drinks and fast food. Adolescents who consumed soft drinks or fast food exhibited a higher prevalence of anxiety-related sleep disturbances than their counterparts who were not involved in such consumption. Table 3 presents the disparities in anxiety-related sleep disturbances across various covariates. Adolescents aged 16 to 18 years exhibited a significantly higher prevalence of anxiety-related sleep disturbances compared to those aged 11 to 15 years (10.5% vs. 2.8%; p = 0.046). Anxiety-related sleep disturbance was significantly more prevalent among adolescents experiencing hunger (8.0% vs. 3.0%; p = 0.003). Students experiencing bullying exhibited a notably higher prevalence of anxiety-related sleep disturbance compared to their non-bullied counterparts (7.4% vs. 2.5%, p = 0.007). Students engaged in physical violence exhibited a higher prevalence of sleep deprivation linked to anxiety (5.6% vs. 1.8%; p = 0.016) (Table 3).

Table 4 illustrates the correlation between the consumption of soft drinks and fast food and the occurrence of anxiety-related sleep disturbance. While the bivariate relationships between fast food or soft drink consumption and anxiety-related sleep disturbance were not significant, multivariable analysis indicated significance. Adolescents attending school who consumed

**Table 2. Sample characteristics (n = 1746).**

| Variables | Number* | Percent (%)* |
|---|---|---|
| **Age** | | |
| 11–15 | 1625 | 90.9 |
| 16–18 | 121 | 9.1 |
| **Gender** | | |
| Male | 666 | 64.1 |
| Female | 1080 | 35.9 |
| **Grade** | | |
| Class VII | 465 | 27.5 |
| Class VIII | 197 | 26.4 |
| Class IX | 888 | 22.5 |
| Class X | 196 | 24.0 |
| **Felt Hungry** | | |
| No | 1553 | 89.9 |
| Yes | 193 | 10.1 |
| **Physical violence** | | |
| No | 1097 | 54.2 |
| Yes | 649 | 45.8 |
| **Physical activity** | | |
| No | 210 | 11.5 |
| Yes | 1536 | 88.5 |
| **Bullied** | | |
| No | 2367 | 75.5 |
| Yes | 608 | 24.5 |
| **Anxiety-related sleep disturbance** | | |
| No | 1685 | 96.5 |
| Yes | 61 | 3.5 |
| **Close Friend** | | |
| No | 129 | 6.8 |
| Yes | 1617 | 93.2 |
| **Peer support** | | |
| No | 1509 | 81.4 |
| Yes | 237 | 18.6 |
| **Parental attachment** | | |
| No | 294 | 21.6 |
| Yes | 1452 | 78.4 |

*Numbers are unweighted and percentage are weighted.

soft drinks more than once daily over the past 30 days exhibited a 2.43-fold increase in the likelihood of experiencing anxiety-related sleep disturbance (odds ratio [OR] = 2.43; 95% confidence interval [CI] = 1.15–5.15), compared to those who either did not consume soft drinks or consumed them less than once daily during the same period. Anxiety-related sleep disturbance was also found to be associated with fast food consumption, with higher odds. Adolescents who consumed fast food on one or more days in the past week were 134% more likely to experience anxiety-related sleep disturbances compared to those who did not consume fast food at all during that period (OR = 2.34; 95% CI = 1.01–5.43). A gradient relationship

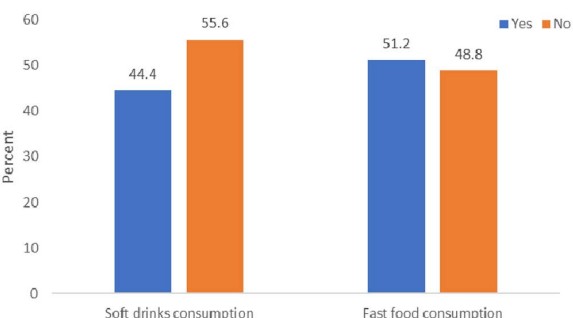

**Fig 1. Prevalence of fast food and carbonated soft drinks among school-going adolescents in Bangladesh.**

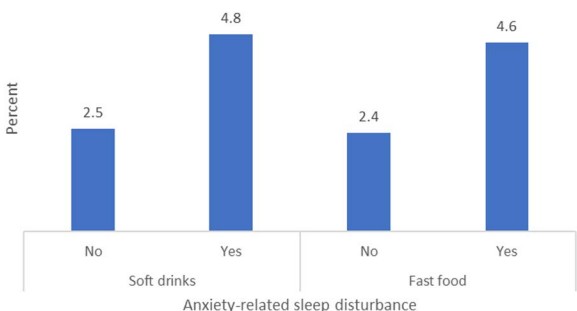

**Fig 2. Anxiety-related sleep disturbances in relation to the consumption of soft drinks and fast food.**

was identified between the consumption of soft drinks and fast food and anxiety-related sleep disturbances. Adolescents who consumed both fast food and soft drinks were 4.39 times more likely (OR = 4.39; 95% CI = 2.02–9.56) to report anxiety-related sleep disturbances compared to those who did not consume these items.

## Discussion

This study evaluated the relationship between carbonated soft drink and fast food intake and the occurrence of sleep deprivation associated with anxiety among a nationally representative sample of school-going adolescents in Bangladesh. The study indicated that adolescents consuming carbonated soft drinks more than once daily over the past month or fast food at least once in the previous week exhibited a higher likelihood of experiencing sleep disturbances attributed to anxiety, even after controlling for other variables. The cross-sectional relationships contribute to the existing literature regarding the links between an unhealthy diet and low well-being among adolescents.

This study, in line with previous global studies [14–17,39,40], demonstrated that the frequency of reporting sleep disturbance caused by anxiety rises as the use of carbonated soft drinks increases. Carbonated beverages frequently include caffeine, a well-known stimulant that has been linked to sleep disturbances in children and adolescents [41, 42]. Moreover, the quantities of caffeine present in carbonated beverages can elicit certain pharmacological and behavioral impacts. Caffeine is recognized for its capacity to improve mood and increase alertness [43, 44]. However, it is important to note that caffeine can also induce feelings of anxiety, irritability, insomnia, and a rapid heartbeat [45–47]. Children who regularly use caffeine may experience restlessness, fidgetiness, headaches, and difficulties falling asleep [46–48]. When

**Table 3. Anxiety-related sleep disturbance by consumption of soft drinks, fast food, and other sociodemographic variables among school-going adolescents; Global School-based Health Survey, Bangladesh, 2014.**

| Variable | Anxiety-related sleep disturbance | | p-value |
|---|---|---|---|
| | Yes, n (%)* | No, n (%)* | |
| **Soft drinks** | | | 0.883 |
| No | 32 (2.5) | 900 (97.5) | |
| Yes | 29 (4.8) | 785 (95.2) | |
| **Fast food** | | | 0.106 |
| No | 19 (2.4) | 734 (97.6) | |
| Yes | 42 (4.6) | 951 (95.4) | |
| **Consumption of soft drinks and fast food** | | | 0.168 |
| Did consume any | 13 (2.0) | 537 (98.0) | |
| Consume at least one | 25 (3.4) | 560 (96.6) | |
| Consume both | 23 (5.4) | 588 (94.6) | |
| **Age group** | | | 0.046 |
| 11–15 years | 53 (2.8) | 1572 (97.2) | |
| 16–18 years | 8 (10.5) | 113 (89.5) | |
| **Gender** | | | 0.569 |
| Male | 15 (3.1) | 651 (96.9) | |
| Female | 46 (4.2) | 1034 (95.8) | |
| **Grade** | | | 0.105 |
| Class VII | 6 (1.0) | 459 (99.0) | |
| Class VIII | 10 (4.7) | 187 (95.3) | |
| Class IX | 37 (3.2) | 851 (96.2) | |
| Class X | 8 (5.4) | 188 (94.6) | |
| **Felt hungry** | | | 0.003 |
| No | 49 (3.0) | 1504 (97.0) | |
| Yes | 12 (8.0) | 181 (92.0) | |
| **Bullied** | | | 0.007 |
| No | 35 (2.5) | 1384 (97.5) | |
| Yes | 26 (7.4) | 301 (92.6) | |
| **Close friends** | | | 0.272 |
| No | 5 (1.9) | 124 (98.1) | |
| Yes | 56 (3.6) | 1561 (96.4) | |
| **Lack of peer support** | | | 0.165 |
| No | 48 (3.0) | 1461 (97.0) | |
| Yes | 13 (6.0) | 224 (94.0) | |
| **Parental attachment** | | | 0.270 |
| No | 21 (5.0) | 273 (95.0) | |
| Yes | 40 (3.1) | 1412 (96.9) | |
| **Physical exercise** | | | 0.103 |
| No | 5 (1.5) | 205 (98.5) | |
| Yes | 56 (3.8) | 1480 (96.2) | |
| **Physical violence** | | | 0.016 |
| No | 25 (1.8) | 1072 (98.2) | |
| Yes | 36 (5.6) | 613 (94.4) | |

*In estimating percentages, complex survey design and sampling weights were taken into account.

**Table 4.** Results from the multivariable logistic regression for the relationship between anxiety-related sleep disturbance and consumption of soft drinks and fast food and other sociodemographic variables among school-going adolescents; Global School-based Health Survey, Bangladesh, 2014.

| Variables | Anxiety-related sleep disturbance | |
|---|---|---|
| | Unadjusted OR (95% CI) | Adjusted OR (95% CI) |
| **Soft Drinks*** | 1.22 (0.72–2.07) | 2.43 (1.15–5.15) |
| **Age group** | | |
| 11–15 years | 1 | 1 |
| 16–18 years | 1.83 (0.80–4.15) | 4.35 (0.74–25.44) |
| **Gender** | | |
| Male | 1 | 1 |
| Female | 2.55 (1.37–4.72) | 2.07 (0.70–6.17) |
| **Grade** | | |
| Class VII | 1 | 1 |
| Class VIII | 3.03 (1.05–8.72) | 3.83 (1.35–10.83) |
| Class IX | 2.88 (1.18–7.03) | 2.40 (0.80–7.60) |
| Class X | 1.99 (0.65–6.09) | 2.54 (0.69–9.29) |
| **Felt hungry** | | |
| No | 1 | 1 |
| Yes | 1.56 (0.79–3.06) | 2.17 (1.19–3.98) |
| **Bullied** | | |
| No | 1 | 1 |
| Yes | 2.66 (1.51–4.68) | 2.53 (1.95–6.73) |
| **Close friends** | | |
| No | 1 | 1 |
| Yes | 1.12 (0.42–3.01) | 2.28 (0.45–10.98) |
| **Lack of peer support** | | |
| No | 1 | 1 |
| Yes | 1.36 (0.68–2.70) | 1.96 (0.51–7.53) |
| **Parental attachment** | | |
| No | 1 | 1 |
| Yes | 0.46 (0.26–0.83) | 0.75 (0.48–2.22) |
| **Physical exercise** | | |
| No | 1 | 1 |
| Yes | 1.46 (0.55–3.86) | 1.84 (0.48–7.06) |
| **Physical violence** | | |
| No | 1 | 1 |
| Yes | 1.69 (0.96–2.96) | 2.16 (0.84–5.59) |
| **Fast food*** | 1.89 (1.05–3.38) | 2.34 (1.01–5.43) |
| **Consumption of soft drinks or fast food*** | | |
| Did not consume any | 1 | 1 |
| Consume least one | 1.47 (0.73–2.98) | 1.49 (0.40–5.53) |
| Consume both | 1.98 (0.95–4.09) | 4.39 (2.02–9.56) |

OR, Odds ratio; 95% CI, 95% confidence interval.

*Models were adjusted age group, gender, school-grade, felt hungry, bullied, close friends, lack of peer support, parental attachment, physical activity, physical violence.

caffeine is consumed at typical levels by humans, the majority of its physiological effects occur through the inhibition of adenosine receptors, specifically the A1 and A2A receptors, and, to a lesser extent, the A2B and A3 receptors. The A1 and A2A adenosine receptors have an impact on several systems found in different regions of the brain that are responsible for regulating sleep, arousal, and cognition [49].

This study demonstrates the relationship between fast food consumption and sleep disturbances linked to anxiety, aligning with previous research [15,16,39,50]. Fast foods are generally characterized by high caloric content and low nutritional value. Frequent consumption of these foods has been associated with various mental health issues [51], including decreased sleep duration [15] and lowered sleep quality among adolescents [52]. Previous studies have demonstrated a relationship between heightened intake of snacks, sugar-sweetened beverages, fast foods, and sedentary behaviors [53]. These behaviors are identified as risk factors for obesity [54]. Obesity in adolescents is recognized as a risk factor for sleep disturbances associated with anxiety [55]. Consumption of fast food may be linked to sleep loss related to anxiety, potentially due to increased inflammation and enhanced immune system activity [56]. Fast food typically contains higher levels of fat, sugar, and energy, which may correlate significantly with factors such as obesity and associated biomarkers [57]. Furthermore, a correlation exists between fast food consumption and social isolation, which may exacerbate the association between fast food consumption and sleep disturbances related to anxiety [58].

Adolescents demonstrate a significantly increased risk for insufficient sleep duration. Multiple specific behavioral risk factors may account for this finding, such as increased electronic device usage and suboptimal dietary habits [59, 60]. A systematic review of studies has shown that the consumption of junk foods, including ultra-processed foods, fast foods, unhealthy snacks, and sugar-sweetened beverages, increases the risk of stress and depression [61]. Sleep disturbances, including insomnia, narcolepsy, sleep-disordered breathing, and restless legs syndrome (RLS), occur in about 90% of individuals with depression [62, 63]. Moreover, adolescence represents a critical transitional phase characterized by elevated stress levels, significant social adjustments, and heightened sensitivity to stress [64]. Research has shown a strong correlation between childhood trauma and current stressful life events with sleep disruptions, even when controlling for depression and anxiety [65].

The increasing prevalence and accessibility of social media in developing nations like Bangladesh, especially among the youth, presents notable concerns regarding its frequent utilization by adolescents. The increase in nighttime use of electronic devices, late-night social activities, substance consumption, fast food intake, and caffeinated drinks, along with early school start times, leads to a greater prevalence of insufficient sleep or reduced sleep quality among adolescents [66]. This usage also correlates with anxiety, stress, depression, reduced mood, and low self-esteem, along with decreased sleep duration and increased sleep disturbances in this population [67]. Furthermore, adolescence is a critical developmental stage where individuals gain autonomy, enhance social skills, and develop a strong tendency to adopt modern lifestyles. The demands placed by educational institutions for increased study commitments and homework significantly reduce sleep duration, resulting in fatigue, fewer opportunities for physical activity, and a greater reliance on energy-dense foods during prolonged wakefulness among young individuals [68]. These social and behavioral factors considerably exacerbate previously hypothesized processes within the body, such as hormone dysregulation, metabolic pathway changes, and daytime drowsiness and weariness [69, 70], which therefore exacerbate problems with sleep. Additionally, adolescents often represent a primary demographic for the marketing of energy-dense, nutrient-poor food and beverage products [71]. An obesogenic environment in adolescents may lead to the development of eating behaviors and dietary choices that heighten the risk of sleep disturbances or sleep loss.

Several cautions must be considered when evaluating the study's findings. First of all, because the study depended on self-reported data, it was subject to recall bias and social desirability. Secondly, as this study exclusively comprised adolescents enrolled in school and present during data collection, data from those who had dropped out or were absent was not available, constraining the generalizability of the findings. Third, the anxiety-related sleep disturbance, potentially indicative of anxiety, was assessed using a single item in the questionnaire. Furthermore, this single-item survey question did not provide data on sleep efficiency, duration, or onset latency, hindering a comprehensive assessment of sleep amount and quality. Fourth, the assessment of fast food and carbonated soft drinks was conducted exclusively on frequency, which may not adequately represent quantity. Fifth, by employing a cross-sectional methodology, we are only able to propose associations between soft drink or fast food intake and anxiety-related sleep disturbance among Bangladeshi adolescents enrolled in school; we are unable to establish causal linkages. Finally, the most recent GSHS in Bangladesh was completed in 2014; hence we lack new evidence regarding the correlation between unhealthy food patterns and sleep disturbances. Evidence indicates a rising consumption of fast food and soft drinks among youth [19,25], and its correlation with anxiety-related sleep disturbances aligns with prior global research [15,16,39,40,50], underscoring the significance of this study's findings.

In summary, this study investigates the correlation between the consumption of carbonated soft drinks and fast food and the occurrence of anxiety-related sleep disturbances in a nationally representative cohort of school-aged adolescents. Our investigation revealed a noteworthy association between the intake of soft beverages or fast food and the manifestation of sleep disturbances attributed to anxiety. Moreover, the consumption of both soft drinks and fast food was linked to a notable rise in the probability of encountering anxiety-related sleep disturbances. While the directionality of the relationship remains unknown, the results of this study indicate the potential benefits of implementing educational interventions for both parents and children via community- and school-based initiatives aimed at addressing unhealthy food consumption and its negative repercussions. Future investigations ought to delve into the relationship through longitudinal methodologies and consider the exploration of potential mediators. This offers a profound understanding of strategies for fostering healthy lifestyles among adolescents in Bangladesh and across the globe.

## Acknowledgements

We would like to thank the World Health Organization for making the Bangladesh Global School-based Health Survey data publicly available.

## Author contributions

**Conceptualization:** Raufun Hasan Arnob, Shamima Akter.

**Data curation:** Raufun Hasan Arnob, Shamima Akter.

**Formal analysis:** Raufun Hasan Arnob, Shamima Akter.

**Methodology:** Md. Mosfequr Rahman.

**Software:** Raufun Hasan Arnob.

**Supervision:** Md. Mosfequr Rahman.

**Writing – original draft:** Raufun Hasan Arnob, Shamima Akter.

**Writing – review & editing:** Md. Mosfequr Rahman.

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
