## [Decision Letter · Decision Letter 0]

14 Oct 2024

PGPH-D-24-01105

Consumption of fast food and carbonated soft drinks, and anxiety-related sleep loss among school-going adolescents in Bangladesh

Dear Dr. Rahman,

Thank you for submitting your manuscript to PLOS Global Public Health. After careful consideration, we feel that it has merit but does not fully meet PLOS Global Public Health’s publication criteria as it currently stands. Therefore, we invite you to submit a revised version of the manuscript that addresses the points raised during the review process.

We look forward to receiving your revised manuscript.

Kind regards,

Feten Fekih-Romdhane

Academic Editor

Journal Requirements:

Additional Editor Comments (if provided):

Reviewers' comments:

Reviewer's Responses to Questions

**Comments to the Author**

1. Does this manuscript meet PLOS Global Public Health’s publication criteria ? Is the manuscript technically sound, and do the data support the conclusions? The manuscript must describe methodologically and ethically rigorous research with conclusions that are appropriately drawn based on the data presented.

Reviewer #1: Partly

Reviewer #2: Partly

Reviewer #3: Yes

Reviewer #4: Yes

2. Has the statistical analysis been performed appropriately and rigorously?

Reviewer #1: I don't know

Reviewer #2: Yes

Reviewer #3: Yes

Reviewer #4: Yes

3. Have the authors made all data underlying the findings in their manuscript fully available (please refer to the Data Availability Statement at the start of the manuscript PDF file)?

Reviewer #1: Yes

Reviewer #2: Yes

Reviewer #3: Yes

Reviewer #4: Yes

4. Is the manuscript presented in an intelligible fashion and written in standard English?

Reviewer #1: Yes

Reviewer #2: No

Reviewer #3: Yes

Reviewer #4: Yes

5. Review Comments to the Author

Reviewer #1: The paper is well written in an easily comprehensible manner. However, I have the following concerns:

1. Given that the data upon which your discussion is based is a decade old, what limitations might this pose to the study? Your response to this could be included in the caveat section that you already discussed in the paper.

2. The p-values in your Table 3 shows that the association of many of your independent variables (including the soft drinks and fast food variables) with the dependent variable (anxiety related sleep loss) is not statistically significant. How may this affect the interpretation of the associations you have provided in the results section?

Reviewer #2: General comments

1)The authors have selected an interesting topic pertaining to adolescent health. Here are few recommendations.

2) The article requires english grammar correction, as few sentences were difficult to comprehend.

Abstract:

In the results section of abstract it would be better to not comment on bullying victimization, substance use and other factors on the effect of sleep.As the main objective focuses on association between consumption of carbonated drinks, fast food on anxiety related sleep loss

Introduction:

1) The authors can provide global statistics pertaining to anxiety related sleep loss among adolescents.

2) As authors have also find association between various factors and anxiety related sleep loss it would be better to add it as secondary objective in the concluding section of introduction.

Methodology:

1)In the first line citing GSHS, the authors can provide reference for the survey.

2) Authors have to mention the study period of secondary data analysis of GSHS data.

Results:

1) The authors can omit Table 1.

Discussion:

1) Discussion looks fragmented. It would be better for the authors to write discussion in sequence aligned with results presentation.

Reviewer #3: The study addresses a pertinent issue in adolescent health—dietary habits and their impact on mental health and sleep quality, which is highly relevant, especially in developing countries like Bangladesh where these trends are under-researched. The manuscript is well-structured, with a clear flow from the abstract to conclusions.

However, there are some minor changes that can be made to this manuscript.

The abstract can be made more concise by shortening the sentences that discuss odds ratios (ORs) could be shortened. Also, the prevalence of hunger, bullying, and other factors that were included could be briefly mentioned without going into too much detail.

In the introduction, the manuscript could benefit from a more focused research question at the end of the introduction, explicitly stating what gap this study addresses in the context of Bangladesh.

In the methodology, clarify some of the operational variables such as "food insecurity" and "physical violence" which can be described in detail or in reference to previous studies that have used these specific measures.

The results section is well-explained, but some parts are over-detailed. For example, the specific prevalence rates of hunger and other variables in Table 2 could be streamlined or moved to supplementary material. It might also help to include a more detailed breakdown of how fast food and soft drink consumption vary by demographic characteristics like age and gender.

The discussion effectively compares findings to previous studies but could be enhanced by integrating a more thorough exploration of cultural or socioeconomic factors unique to Bangladesh that might explain the results.

The discussion on the potential biological mechanisms of how fast food and soft drinks might contribute to anxiety-related sleep is valuable but should also consider psychosocial mechanisms such as stress related to unhealthy dietary habits.

Reviewer #4: - In the title, specify "anxiety-related sleep loss" to "sleep disturbances associated with anxiety" for clarity.

- For abstract, mention the covariates used in the analysis to provide a more complete overview of the study's methodology.

- In the introduction, strengthen the transition between global findings and the Bangladesh-specific context, particularly including lifestyle changes due to urbanization.

- Methods section, justify the inclusion of covariates in the logistic regression model by explaining their connection to anxiety-related sleep loss. Clarify how missing data were handled, what tests were done. Further explain why certain variables, such as substance use and physical violence, were included and how they were measured.

- Visuals.. for results, add visual representations (e.g., bar graphs) to show the distribution of fast food and soft drink consumption in relation to sleep loss for better clarity.

- In the results, provide a clearer explanation of how confounding variables influence the relationship between fast food, soft drinks, and anxiety-related sleep loss.

- In the discussion, consider reverse causality—whether sleep loss leads to increased consumption of fast food and soft drinks. Also, expand on the limitations, particularly regarding the biases from self-reported data on sensitive topics like substance use and sexual behavior.

- The conclusion is vague right now, include specific recommendations for policymakers and public health practitioners in Bangladesh to apply the findings.

- In tables and figures, clarify labels and legends, especially for Figure 1, to improve data interpretation.

- In the references, ensure that all references are directly relevant to the findings and reduce reliance on studies from high-income countries without proper justification.

6. PLOS authors have the option to publish the peer review history of their article (what does this mean? ). If published, this will include your full peer review and any attached files.

**Do you want your identity to be public for this peer review?** For information about this choice, including consent withdrawal, please see our Privacy Policy .

Reviewer #1: No

Reviewer #2: No

Reviewer #3: No

Reviewer #4: No

---

## [Decision Letter · Decision Letter 1]

21 Jan 2025

PGPH-D-24-01105R1

Consumption of carbonated soft drinks and fast food, and sleep disturbance associated with anxiety among school-going adolescents in Bangladesh

Dear Dr. Rahman,

Thank you for submitting your manuscript to PLOS Global Public Health. After careful consideration, we feel that it has merit but does not fully meet PLOS Global Public Health’s publication criteria as it currently stands. Therefore, we invite you to submit a revised version of the manuscript that addresses the points raised during the review process.

We look forward to receiving your revised manuscript.

Kind regards,

Feten Fekih-Romdhane

Academic Editor

Journal Requirements:

Additional Editor Comments (if provided):

Before we can accept your paper for publication, the title should be changed to : "Evaluation of the relationship between consumption of carbonated soft drinks/fast food and anxiety-related sleep disturbance in school adolescents in Bangladesh."

Reviewers' comments:

Reviewer's Responses to Questions

**Comments to the Author**

1. If the authors have adequately addressed your comments raised in a previous round of review and you feel that this manuscript is now acceptable for publication, you may indicate that here to bypass the “Comments to the Author” section, enter your conflict of interest statement in the “Confidential to Editor” section, and submit your "Accept" recommendation.

Reviewer #1: All comments have been addressed

Reviewer #2: All comments have been addressed

Reviewer #3: All comments have been addressed

Reviewer #4: All comments have been addressed

2. Does this manuscript meet PLOS Global Public Health’s publication criteria ? Is the manuscript technically sound, and do the data support the conclusions? The manuscript must describe methodologically and ethically rigorous research with conclusions that are appropriately drawn based on the data presented.

Reviewer #1: (No Response)

Reviewer #2: Yes

Reviewer #3: Yes

Reviewer #4: Yes

3. Has the statistical analysis been performed appropriately and rigorously?

Reviewer #1: (No Response)

Reviewer #2: Yes

Reviewer #3: Yes

Reviewer #4: Yes

4. Have the authors made all data underlying the findings in their manuscript fully available (please refer to the Data Availability Statement at the start of the manuscript PDF file)?

Reviewer #1: (No Response)

Reviewer #2: Yes

Reviewer #3: Yes

Reviewer #4: Yes

5. Is the manuscript presented in an intelligible fashion and written in standard English?

Reviewer #1: (No Response)

Reviewer #2: Yes

Reviewer #3: Yes

Reviewer #4: Yes

6. Review Comments to the Author

Reviewer #1: (No Response)

Reviewer #2: The authors have addressed all the comments.

Reviewer #3: (No Response)

Reviewer #4: All comments have been addressed.

7. PLOS authors have the option to publish the peer review history of their article (what does this mean? ). If published, this will include your full peer review and any attached files.

**Do you want your identity to be public for this peer review?** For information about this choice, including consent withdrawal, please see our Privacy Policy .

Reviewer #1: No

Reviewer #2: No

Reviewer #3: No

Reviewer #4: No

---

## [Editor Report · Decision Letter 2]

4 Feb 2025

Evaluation of the relationship between consumption of carbonated soft drinks/fast food and anxiety-related sleep disturbance in school adolescents in Bangladesh

PGPH-D-24-01105R2

Dear Prof. Rahman,

We are pleased to inform you that your manuscript 'Evaluation of the relationship between consumption of carbonated soft drinks/fast food and anxiety-related sleep disturbance in school adolescents in Bangladesh' has been provisionally accepted for publication in PLOS Global Public Health.

Best regards,

Feten Fekih-Romdhane

Academic Editor